# Differential risk of cardiovascular complications in patients with type-2 diabetes mellitus in Ghana: A hospital-based cross-sectional study

**Christian Obirikorang**[1,2]*, **Evans Asamoah Adu**[1,2], **Anthony Afum-Adjei Awuah**[1,2], **Samuel Nkansah Darko**[1], **Frank Naku Ghartey**[3], **Samuel Ametepe**[1,4], **Eric N. Y. Nyarko**[5], **Enoch Odame Anto**[6,7], **William Kwame Boakye Ansah Owiredu**[1]

1 Department of Molecular Medicine, School of Medical Sciences, Kwame Nkrumah University of Science and Technology (KNUST), Kumasi, Ghana, 2 Global Health and Infectious Disease, Kumasi Centre for Collaborative Research in Tropical Medicine, Kumasi, Ghana, 3 Department of Chemical Pathology, School of Medical Sciences, University of Cape Coast, Cape Coast, Ghana, 4 Department of Medical Laboratory Sciences, Koforidua Technical University, Koforidua, Ghana, 5 Department of Chemical Pathology, University of Ghana Medical School, University of Ghana, Accra, Ghana, 6 Department of Medical Diagnostics, Faculty of Allied Health Sciences, Kwame Nkrumah University of Science and Technology, Kumasi, Ghana, 7 Centre for Precision Health, School of Medical and Health Sciences, Edith Cowan University, Joondalup, Western Australia, Australia

* krisobiri@yahoo.com

## Abstract

### Aim

To characterize clinically relevant subgroups of patients with type-2 diabetes mellitus (T2DM) based on adiposity, insulin secretion, and resistance indices.

### Methods

A cross-sectional study was conducted at Eastern Regional Hospital in Ghana from July to October 2021 to investigate long-term patients with T2DM. To select participants, a systematic random sampling method was employed. Demographic data was collected using a structured questionnaire and fasting blood samples were taken to measure glycemic and lipid levels. Blood pressure and adiposity indices were measured during recruitment. The risk of cardiovascular disease (CVD) was defined using Framingham scores and standard low-density lipoprotein thresholds. To analyze the data, k-means clustering algorithms and regression analysis were used.

### Results

The study identified three groups in female patients according to body mass index, relative fat mass, glycated hemoglobin, and triglyceride-glucose index. These groups included the obesity-related phenotype, the severe insulin resistance phenotype, and the normal weight phenotype with improved insulin resistance. Among male patients with T2DM, two groups were identified, including the obesity-related phenotype with severe insulin resistance and

---

**Data Availability Statement:** The dataset that supports the findings of this paper and associated

---

codes for analysis can be accessed from figshare at DOI:10.6084/m9.figshare.27610797 (https://doi.org/10.6084/m9.figshare.27610797.v1).

**Funding:** The author(s) received no specific funding for this work.

**Competing interests:** The authors have declared that no competing interests exist.

the normal weight phenotype with improved insulin sensitivity. The severe insulin resistance phenotype in female patients was associated with an increased risk of high CVD (OR = 5.34, 95%CI:2.11–13.55) and metabolic syndrome (OR = 7.07; 95%CI:3.24–15.42). Among male patients, the obesity-related phenotype with severe insulin resistance was associated with an increased intermediate (OR = 21.78, 95%CI:4.17–113.78) and a high-risk CVD (OR = 6.84, 95%CI:1.45–32.12).

## Conclusions

The findings highlight significant cardiometabolic heterogeneity among T2DM patients. The subgroups of T2DM patients characterized by obesity and/or severe insulin resistance with or without poor glycemic control, have increased risk of CVD. This underscores the importance of considering differences in adiposity, insulin secretion, and sensitivity indices when making clinical decisions for patients with T2DM.

## Introduction

Type-2 diabetes mellitus (T2DM) is a chronic condition characterized by high levels of circulating blood glucose. It is the most common type of diabetes, typically affecting adults with or without obesity-associated complications [1]. While a diagnosis of diabetes can be made based on glycated hemoglobin and/or fasting blood glucose levels [1, 2], this does not provide a definitive classification of the disease. Thus, T2DM diagnosis is made in the absence of type-1 diabetes mellitus, which has unique immune and genetic markers for classification [2]. Unfortunately, the current approach to classifying T2DM falls short in understanding the epidemiological and clinical landscape of the disease. Moreover, multiple risk factors can influence the progression of T2DM, making it difficult to predict prognosis and track disease progression [3].

The prevalence of diabetes has increased worldwide, especially in low and middle-income countries [1]. In the International Diabetes Federation (IDF) report, 1 in 10 adults have diabetes worldwide, with 3 in 4 adults in low- and middle-income countries [4]. In Africa, diabetes is projected to increase by 129% by 2045, with mortality rates increasing by up to 20.8% between 2011 and 2021 [4]. In Ghana, 2.6% of the adult population is estimated to have diabetes, which is projected to increase to 3.3% by 2045 [4]. However, aggregated data from observational studies suggest even higher estimates at 6.5 (4.7–8.3)% [5]. Thus, the impact of diabetes is felt globally, affecting social, economic, and personal livelihoods [6, 7]. According to the Global Health estimate [8], T2DM is a major cause of disability and death.

The above text sheds light on the mounting responsibility of managing diabetes and the shortcomings of the current standard definition. In Ghana, the majority of individuals with T2DM have poor control [9], and hospitalizations are on the rise annually [10, 11]. A large part of this can be attributed to the current clinical definition, which fails to link diagnosis with underlying pathophysiological factors. Consequently, achieving optimal metabolic control and identifying those who require intensified treatment is a challenge [12–15]. Additionally, the current treatment approach for T2DM is dependent on the availability and affordability of laboratory tests and medications at the time of patient presentation [1, 12]. These limitations have a significant impact on the effectiveness of existing treatment guidelines and contribute to increased morbidity and mortality rates associated with T2DM [9, 16].

Recent studies suggest that personalized treatment strategies for T2DM patients could be more effective in preventing cardiovascular complications [17–19]. To achieve this, it is crucial to efficiently analyze patient data to identify metabolic differences [15, 18]. There is a considerable variance in the T2DM population, which is why new classification algorithms based on clinical and genetic data are being proposed [3, 13, 19–21]. For example, Ahlqvist *et al*., [13] discovered five groups of diabetic patients with varying risks of complications through unsupervised grouping of patient data, and these findings have been replicated in other studies [22–24]. This demonstrates that unsupervised learning models have the potential to pinpoint clinically relevant subgroups of T2DM patients.

The primary objective of this study was to use unsupervised data-driven cluster analysis to identify subgroups of long-term T2DM patients who were receiving treatment at a tertiary hospital in Ghana. Thus, providing baseline evidence of clinically relevant T2DM subgroups that could influence future large-scale research. We based our clustering analysis on indices that reflect inherent pathophysiology in T2DM such as obesity, insulin resistance, and poor glycemic control.

## Materials and methods

### Study setting and design

According to recent statistics, hospitalization trends among patients with diabetes were markedly high in the Eastern compared to the other regions in Ghana [10]. Patients were recruited from the Eastern Regional Hospital, Koforidua (ERHK) in Ghana from July to October 2021 for this study. The diabetes clinic is held twice a week with an average patient attendance of 140 patients per week [25].

### Sample design and population

The study design was a hospital-based cross-sectional study. The sampling frame was patients with established T2DM, on treatment (between 1 year and 30 years) and aged 30 to 70 years. We emphasized our sample size calculation on the proposed existence of T2DM subpopulations with differential cardiometabolic risk profiles. According to previous studies [15, 22, 26] the proportion of the minimum subgroup of the T2DM population ranges between 5% and 21%. Based on these estimates, we assumed a maximum permissible limit of a minimum subgroup prevalence of 21%. Hence, we chose a relative precision of (21/100) *21 = 4.4% for our sample size calculation. This means that the sample size represents a minimum prevalence of 18.8% ± 2.2% (18.8% to 23.2%). At a 95% confidence interval, and based on the above assumptions, the minimum size of T2DM cases required to detect a proportion of 18.8% to 23.2% of the minimum group was 330 patients. Our calculation was based on the following formulae:

$n = \frac{z^2 \times p(1-p)}{e^2}$ [27]. Where n = sample size, z = 1.96, p = 21%, e = 4.4%.

### Inclusion and exclusion criteria

Patients were included in the study if they met the following criteria: confirmed clinically diagnosed patients with T2DM (ICD-10 code: E11) who have been treated for at least one year, between 30 and 70 years old, do not have comorbid hypertension or a history of cardiovascular disease, including stroke, myocardial infarction, and coronary artery disease. Patients with T2DM who self-managed or were being treated for microvascular complications were excluded from the study. Additionally, patients whose medical records revealed a probable alteration of the metabolic profile due to conditions including peripheral vascular disease

(PVD), inflammatory and autoimmune disorders such as rheumatoid arthritis (RA), and other indications of corticosteroid therapy were excluded from this study.

## Sampling and recruitment

Patient recruitment was integrated into routine clinic visits for patients with diabetes at ERHK. We considered that if an average of 140 patients visits per week [25], then 70 diabetes patients are expected for each clinic visit day. Based on this information, we adopted a systematic random sampling approach for patient recruitment on a first-come-first-chosen basis within the 4-month (16 weeks) time window. After the first patient was selected, an estimated sample interval was calculated based on the expected patients per clinic visit: 70 patients by 2 clinic visits by 16 weeks, divided by our estimated sample size (n = 330). That is, every seventh patient was selected after the first patient is recruited on a particular visit day (approximately, 10 patients were sampled per clinic visit). Whenever a patient refused or withdraws consent or did not meet the eligibility criteria, the next available patient was recruited.

## Data and sample collection

A structured questionnaire was used for data collection. The questionnaire had sections for demographic data for patients (current age, sex, family history of DM, past and current history of smoking and alcohol intake), clinical history (age at the time of diagnosis of diabetes, duration of treatment, symptoms, diagnostic information on complication and medication regime), anthropometric measurements and laboratory tests. It is routine practice for patients to fast overnight before visiting the clinic for health examinations. Thus, laboratory measurements are taken in fasting state.

## Anthropometry and blood pressure measurements

For each recruited participant, height and weight were measured using a stadiometer and an electronic weigh balance whiles in light cloth (OMRON HEALTHCARE Co., Ltd.), respectively. Waist circumference (WC) was measured at the umbilicus level and the maximum gluteal protrusion using a tape measure. BMI was calculated as body weight/height (kg/m$^2$). RFM was calculated from WC and height:

$$RFM = (64 - (20*(height/waist\ circumference) + (12 \times ifelse\ (sex == female, 1, 0)$$

Blood pressure was checked by trained staff nurses and recovered from the patient's folder. In the laboratory, blood samples (2 millilitres in sodium fluoride tube and 3 millilitres in serum separator tubes) were drawn, processed, and stored (-20˚C) for subsequent measurements. All measurement were performed whiles patients were in the fasting state.

## Laboratory measurements

Laboratory analysis of the samples was performed at the ERHK Clinical Chemistry Laboratory. Sodium fluoride whole blood samples were used to estimate HbA1c with a standard A1c Care System (SD BIOSENSOR, Kyonggi-do, Korea). Fasting plasma glucose (FPG) and lipids were estimated using plasma and serum obtained from sodium fluoride and serum separator tubes, respectively, on the SELECTRAPRO M chemistry analyzer (EliTech Group B.V, The Netherlands). We recorded glucose and lipid concentrations in mmol/L and HbA1c as percentages. The lipid profile included high-density lipoprotein cholesterol (HDL-c), low-density lipoprotein cholesterol (LDL-c), triglycerides (Trig), and total cholesterol (T. Chol). Non-HDL-c was estimated as the difference between T. Chol and HDL-c.

## Definition of outcome variables

CVD risk outcomes were defined using the following scores and definitions:

- The updated 2008 Framingham Risk Score for the risk of general Atherosclerotic cardiovascular disease (ASCVD) and the risk of individual CVD events (coronary, cerebrovascular, and peripheral arterial disease, and heart failure) was used to define the 10-year risk of heart diseases [28].

- We adopted the criteria of the American Diabetes Association (ADA) criteria [29] and the 2013 American College of Cardiology (ACC) and the American Heart Association (AHA) guidelines [30] to categorize low-density lipoprotein cholesterol: < 3.4 mmol/L (near optimal); 3.4–4.1 mmol/L (borderline high); ≥ 4.2 mmol / L (high).

Based on the above categories, we defined CVD risk phenotypes on an ordinal scale using **Table 1** below.

## Definitions of other outcome variables

General obesity was defined based on BMI thresholds for overweight ($\geq$25 Kg/m$^2$), and obesity ($\geq$30 Kg/m$^2$) according to the World Health Organization's criteria [31]. Central obesity was defined using sex and age-specific cut-off values for RFM as described in a previous study [32]:

- Males: 20–39 years ≥25; 40–59 years ≥28; 60–79 years >30.

- Females: 20–39 years ≥ 39; 40–59 years ≥40; 60–79 years >42

We created categories of triglyceride and HDL-c levels based on the 2013 American College of Cardiology/American Heart Association guidelines [33]: normal triglyceride (triglyceride <1.7 mmol/L), high triglyceride (triglyceride 1.7 mmol / L), normal HDL-c (HDL-c 1.0 mmol / L for men and 1.3 mmol / L for women), and low HDL-c (HDL-c <1.0 mmol/L for men and <1.3 mmol/L for women). We defined metabolic dyslipidemia as high triglyceride and low HDL according to the definition proposed in a previous study [34].

We adopted the Adult Treatment Panel III (ATP III) of the National Cholesterol Education Program (NCEP) Adult Treatment Panel III (ATP III) [35] for the definition of Metabolic syndrome (MetS). That is, if two of the following criteria are met in addition to the presence of T2DM: waist circumference <102 cm (men) or <88 cm (women), blood pressure <130/85 and dyslipidaemia (fasting triglyceride level >1.7 mmol/L and/or HDL-c less than 1.03 mmol/l (men) or 1.29 mmol/l (women). Severe insulin resistance was defined as TyG levels ≥ 9.0 for women and ≥9.2 for men [36]. Poor glycemic control was defined according to the recommendations of the American Diabetes Association: HbA1c >7.0% for both men and women [29].

**Table 1. Adapted definition for cardiovascular risk among patients with T2DM.**

| LDL-c categories | ASCVD risk scores | | | Key | |
|---|---|---|---|---|---|
| | **Low risk** | **Intermediate risk** | **High risk** | **Low** | |
| Optimal | | | | **Intermediate** | |
| Borderline high | | | | **High** | |
| High | | | | | |

LDL-c: < 3.4 mmol/L (Optimal); 3.4–4.1 mmol/L (Borderline high); ≥ 4.2–4.9 mmol / L (high). ASCVD risk scores: <10% (low risk); 10–19.9% (intermediate risk); 20% (high risk).

## Cluster analysis

**Assumption.** Variables for the clustering model were selected on the basis that the pathophysiological progression to T2DM includes obesity, metabolic syndrome, and glucose dysregulation, due to systemic insulin resistance [37, 38]. The combined impact of these abnormalities imposes glucolipotoxicity leading to eventual β-cell decline with decreased insulin secretion [37]. At this point, the development of cardiovascular complications is inevitable [39]. Thus, we have used BMI and RFM as proxies for central and general obesity, HbA1c for glucose control and β-cell function, and TyG for systemic insulin resistance.

**Variables.** The selected indices included Body Mass Index (BMI), Relative Fat Mass (RFM), Glycated Hemoglobin (HbA1c), and Triglyceride-Glucose Index (TyG). BMI is a well-characterized and most used adiposity index in daily clinical work in patients with T2DM. Despite the obesity paradox in patients with T2DM [40, 41], elevated BMI has widely been replicated in several studies as a risk factor for both the onset and progression of T2DM [42–45]. The RFM represents a newly proposed adiposity index that overcomes the limitations of BMI by accurately defining obesity defined by body fat-defined obesity [46–48]. In several studies, RFM has been shown to be effective for use in the general practice setting to estimate the risk of both the onset and progression of T2DM [49, 50]. The TyG index is a simple method for determining insulin resistance [51]. Some studies reported that the TyG index is superior to the Homeostatic Model Assessment for Insulin Resistance (HOMA-IR) in predicting T2DM [52, 53]. Moreover, TyG has been used for predicting a variety of CVD types [54–59]. HbA1c is an important indicator of long-term glycemic control that is a reliable biomarker in routine practice for the diagnosis and prognosis of diabetes [60]. Therefore, using these indices together in a cluster analysis provides a cumulative cardiometabolic history of the population with T2DM.

**Clustering.** We performed the cluster analysis for men and women data separately to avoid any stratification due to sex-dependent differences. We tested several methods for an optimal number of clusters (**S1 File**) and performed internal validation analysis using the average Silhouette width, Dunn index, and connectivity index. We used K-means analysis according to the Hartigan and Wong algorithm [61] to cluster the data. Clustering analysis was performed with the 'NbClust' package [62] on R software (Version 4.3.3). The details of the clustering analysis can be found in the S1 File. The characteristics of the variables used for the clustering is shown in **S1 File** (**S1 Fig**). S2 and S3 Figs shows the optimal number of clusters that exist in male and female.

## Data management and statistical analysis

All data obtained from the patients were managed on Microsoft Spreadsheet: double checked after entry for precision and reliability and locked before usage. The data required for analysis were extracted in a CSV file. Data analysis was performed using the R software (Version 4.3.3) and the Statistical Package for Social Sciences (SPSS version 25). Categorical data were presented as counts and corresponding percentages. The chi-square test was used to compare data differences in categorical variables between the T2DM subgroups. Summary statistics of continuous variables were presented as mean and standard deviations (for normally distributed data) and compared using a t-test and one-way analysis of variance when required. Skewed data was presented as median and interquartile ranges and differences between subgroups computed with Mann-Whitney or Kruskal Wallis test. We used alluvial diagrams to illustrate the flow of cardiometabolic risk factors within each T2DM group. To understand the impact of variables on the existence of clinical subgroups and cardiovascular risk outcomes,

we used multiple regression models with a 'logit' link and reported p-values less than 0.05 as statistically significant.

## Regulatory and ethical approval

The study was carried out according to the ethical principles of the Declaration of Helsinki and the applicable national ethical regulatory requirements. The Human Research, Publication and Ethics Committee (CHRPE) of Kwame Nkrumah University of Science and Technology (KNUST) granted ethical clearance (Ref: CHRPE/AP/217/21) for this study. Additionally, permission was requested and received from the ERHK to use their medical resources for the recruitment of research participants. Under the CHRPE study participant information policies, written information was provided to participants and their consent was obtained prior to admission to the study. Participation was completely voluntary and all patients with T2DM received equal participation rights.

## Results

### Demographic, clinical, and laboratory characteristics of female and male T2DM groups

The characteristics of male T2DM clusters are shown in **Table 2**. Two (2) clusters were obtained for male T2DM data. Cluster 1 constituted 62.1% of the male data and cluster 2 constituted 37.9%. Clusters 1 and 2 did not differ in terms of their demographic, medication prescription, and symptom history. Regarding laboratory profile, Cluster 2 had significantly higher levels of triglycerides (p-value = 0.001), total cholesterol (p-value = 0.006) and non-HDL-c (p-value = 0.022) than Cluster 1 (**Table 2**). The waist circumference was elevated in group 2 than in group 1 (p-value <0.001, **Table 2**).

The cluster analysis resulted in three distinct clusters within the female data from T2DM: 36.1% for cluster 1 and 31.9% each for clusters 2 and 3 (**Table 3**). Sociodemographic background did not differ between the three clusters (p-value >0.05). A significantly higher proportion of cluster 2 than cluster 1 had a history of taking Metformin (98.6% vs. 84.2%), while a significantly higher proportion of cluster 1 than cluster 2 (13.2% vs. 2.9%) took Thiazolidinedione (p-value <0.05). Furthermore, a higher proportion of patients in Group 2 and 3 than in Group 1 reported symptoms of easy fatiguability (p-value = 0.03, **Table 3**). In terms of its laboratory profile, cluster 2 had significantly higher fasting blood glucose (p-value <0.001), total cholesterol (p-value < 0.001), non-HDL-c (p-value < 0.001) and diastolic blood pressure (p-value = 0.004) than clusters 1 and 3. All three groups were distinct in terms of their mean triglyceride (p-value < 0.001) and waist circumference (p-value < 0.001). Clusters 2 and 3 had significantly elevated mean LDL-c compared to cluster 1 (p-value <0.001).

### Features of the female and male T2DM clusters

As shown in **Fig 1A,** the three groups were distinct in terms of their median levels of BMI, RFM, HbA1c and TyG. Clusters 2 and 3 had characteristically distinct elevated median lipid accumulation product index compared to Cluster 1 (p-value <0.01). Cluster 2 had a significantly elevated median coronary and atherogenic risk score than Cluster 1 but was comparable to Cluster 3.

As shown in **Fig 1B**, Cluster 2 from male T2DM data had distinctively high BMI and RFM compared to Cluster 1 (p-value <0.001). Again, the lipid accumulation product index and TyG for cluster 2 were significantly elevated compared to cluster 1 (p-value <0.001). However,

**Table 2.  Demographic, clinical, anthropometric, and laboratory characteristics of male subgroups of T2DM.**

| Variable | Male (n = 103) | | |
|---|---|---|---|
| | Cluster 1 N = 64(62.1%) | Cluster 2 N = 39(37.9%) | p-value |
| **Education** | | | 0.092 |
| Non-formal | 5 (7.8) | 0 | |
| Basic | 19 (29.7) | 7 (17.9) | |
| Secondary | 24 (37.5) | 16 (41.0) | |
| Tertiary | 16 (25.0) | 16 (41.0) | |
| **Family history of DM** | | | |
| No | 29 (45.3) | 14 (35.9) | |
| Yes | 35 (54.7) | 25 (64.1) | 0.347 |
| **Currently smoking** | | | |
| No | 59 (92.2) | 38 (97.4) | |
| Yes | 5 (7.8) | 1 (2.6) | 0.404 |
| **Alcohol intake** | | | |
| No | 49 (76.6) | 32 (84.2) | |
| Yes | 15 (23.4) | 6 (15.8) | 0.451 |
| **Medication history** | | | |
| Metformin | 53 (88.3) | 32 (86.5) | 0.788 |
| Insulin | 28 (46.7) | 14 (37.8) | 0.394 |
| Thiazolidinediones | 1 (1.7) | 2 (5.4) | 0.556 |
| Sulphonyl urea | 50 (83.3) | 34 (91.9) | 0.229 |
| DPP-4 | 2 (3.3) | 3 (8.1) | 0.366 |
| Statins | 16 (25.0) | 16 (41.0) | 0.088 |
| **Symptom history** | | | |
| Easy fatiguability | 8 (12.7) | 2 (5.1) | 0.310 |
| Breathlessness | 1 (1.6) | 0 | n/a |
| Chest pain | 8 (12.5) | 3 (7.7) | 0.527 |
| Palpitations | 11 (17.2) | 2 (5.1) | 0.124 |
| **Metabolic profiles^** | | | |
| FBS (mmol/L) | 8.40±2.96 | 8.96±2.45 | 0.322 |
| Trig (mmol/L) | 0.88±0.34 | 1.26±0.62 | **0.001** |
| HDL-c (mmol/L) | 1.37±0.48 | 1.39±0.37 | 0.861 |
| T. Chol (mmol/L) | 4.59±0.99 | 5.11±1.16 | **0.016** |
| LDL-c (mmol/L) | 3.04±0.86 | 3.45±1.16 | 0.063 |
| Non-HDL-c (mmol/L) | 3.22±0.94 | 3.73±1.27 | **0.022** |
| **Anthropometric indices^** | | | |
| WC (cm) | 83.72±6.40 | 99.29±6.17 | **<0.001** |
| Systolic BP (mmHg) | 119.58±13.32 | 124.77±9.63 | **0.024** |
| Diastolic BP (mmHg) | 71.47±8.68 | 73.44±7.37 | 0.241 |
| Age (years) | 51.77±9.20 | 51.15±8.39 | 0.736 |
| Age at T2DM onset (years) | 44.03±9.26 | 42.95±6.51 | 0.589 |
| Duration of T2DM (years)# | 6.0 (3.0–13.0) | 7.0 (3.0–13.0) | 0.586 |

Values are presented as frequencies and compared using chi-square test, unless otherwise specified. Variable with ^ symbol are presented as mean and standard deviation and compared using student t-test. Variables with # symbol are presented as median (interquartile ranges) and compared using Mann-Whitney test. FBS- Fasting Blood Sugar; Trig- Triglycerides; HDL-C- High density lipoprotein cholesterol; T. Chol- Total Cholesterol; LDL-C- Low density lipoprotein cholesterol; DM- diabetes mellitus; BP- blood pressure; DPP-4- inhibitors of dipeptidyl peptidase 4; WC- waist circumference.

**Table 3. Demographic, clinical, anthropometric, and laboratory characteristics of the female subgroups of T2DM.**

| Variable | Female (n = 238) | | | |
|---|---|---|---|---|
| | Cluster 1 N = 86(36.1%) | Cluster 2 N = 76(31.9%) | Cluster 3 N = 76(31.9%) | P-value |
| **Education** | | | | 0.693 |
| Non-formal | 7 (8.2) | 7 (9.2) | 6 (7.9) | |
| Basic | 33 (38.8) | 24 (31.6) | 24 (31.6) | |
| Secondary | 42 (49.4) | 37 (48.7) | 40 (52.6) | |
| Tertiary | 3 (3.5) | 8 (10.5) | 6 (7.9) | |
| **Family history of DM** | | | | 0.162 |
| No | 39 (45.3) | 25 (32.9) | 25 (32.9) | |
| Yes | 47 (54.7) | 51 (67.1) | 51 (67.1) | |
| **Currently smoking** | | | | |
| No | 86 (100.0) | 76 (100.0) | 76 (100.0) | n/a |
| Yes | 0 | 0 | | |
| **Alcohol intake** | | | | |
| No | 83 (96.5) | 72 (94.7) | 72 (94.7) | |
| Yes | 3 (3.5) | 4 (5.3) | 4 (5.3) | 0.822 |
| **Medication** | | | | |
| Metformin | 64 (84.2)[a] | 68 (98.6)[b] | 65 (92.9)[a,b] | **0.007** |
| Insulin | 32 (42.1) | 20 (29.0) | 20 (28.6) | 0.141 |
| Thiazolidinediones | 10 (13.2)[a] | 2 (2.9)[b] | 6 (8.6)[a,b] | 0.083 |
| Sulphonyl urea | 63 (82.9) | 58 (84.1) | 55 (78.6) | 0.674 |
| DPP-4 | 0 | 2 (2.9) | 0 | n/a |
| Statins | 37 (43.0) | 35 (46.1) | 35 (46.1) | 0.903 |
| **Symptom history** | | | | |
| Easy fatiguability | 11 (12.8)[a] | 19 (25.0)[b] | 22 (28.9)[b] | **0.03** |
| Breathlessness | 5 (6.0) | 6 (8.0) | 4 (5.3) | 0.783 |
| Chest pain | 9 (10.8) | 5 (6.6) | 4 (5.4) | 0.400 |
| Palpitations | 10 (11.6) | 4 (5.3) | 10 (13.2) | 0.227 |
| **Metabolic profiles ^** | | | | |
| FBS (mmol/L) | $7.83 \pm 3.02$[a] | $10.35 \pm 3.16$[b] | $8.05 \pm 2.77$[a] | **<0.001** |
| Trig (mmol/L) | $0.81 \pm 0.30$[a] | $1.49 \pm 0.49$[b] | $1.05 \pm 0.32$[c] | **<0.001** |
| HDL-c (mmol/L) | $1.47 \pm 0.35$ | $1.43 \pm 0.48$ | $1.33 \pm 0.50$ | 0.126 |
| T. Chol (mmol/L) | $4.69 \pm 1.11$[a] | $5.74 \pm 1.42$[b] | $4.72 \pm 1.10$[a] | **<0.001** |
| LDL-c (mmol/L) | $2.92 \pm 0.95$[a] | $3.69 \pm 1.30$[b] | $3.05 \pm 1.00$[b] | **<0.001** |
| Non-HDL-c (mmol/L) | $3.22 \pm 1.05$[a] | $4.31 \pm 1.50$[b] | $3.40 \pm 1.07$[a] | **<0.001** |
| **Anthropometric indices^** | | | | |
| WC (cm) | $86.59 \pm 6.30$[a] | $96.12 \pm 7.50$[b] | $104.39 \pm 9.14$[c] | **<0.001** |
| Systolic BP (mmHg) | $119.44 \pm 11.12$ | $119.39 \pm 10.20$ | $115.71 \pm 11.34$ | 0.052 |
| Diastolic BP (mmHg) | $71.66 \pm 6.44$[a] | $75.08 \pm 7.74$[b] | $71.99 \pm 6.74$[a] | **0.004** |
| Age (years) | $50.85 \pm 8.73$ | $50.28 \pm 7.42$ | $50.12 \pm 7.78$ | 0.829 |
| Age at T2DM onset (years) | $41.81 \pm 8.56$ | $42.41 \pm 6.82$ | $43.03 \pm 7.56$ | 0.612 |
| Duration of T2DM (years)[#] | 7.0 (5.0–12.0) | 6.0 (4.0–12.0) | 6.0 (3.0–10.5) | 0.067 |

Values are presented as frequencies and compared using chi-square test, unless otherwise specified. Variable with ^ symbol are presented as mean and standard deviation and compared using student t-test. Variables with # symbol are presented as median (interquartile ranges) and compared using Mann-Whitney test. DM-diabetes mellitus; BP- blood pressure; DPP-4- inhibitors of dipeptidyl peptidase 4; WC- waist circumference. Superscript alphabets (a, b, c) represent differences between the three cluster according to multiple testing with Bonferroni's correction. Clusters with the same superscript alphabet do not differ significantly.

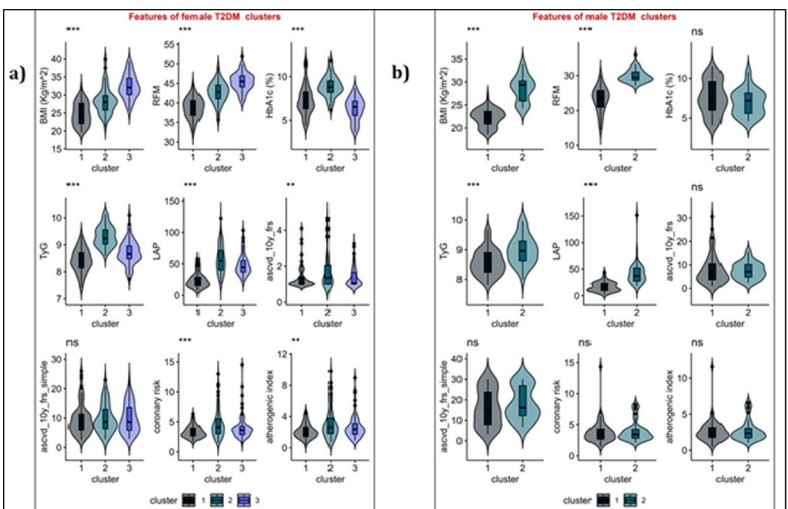

**Fig 1.** Characteristics of the (a) female and (b) male T2DM groups. The Mann-Whitney test was used to compute probability values. Asymptomatic probability values (2-sided tests) are indicated with the * symbol. *** P-value <0.001; ** p-value <0.01, *p-value <0.05, ns p-value >0.05. HbA1c = glycated hemoglobin, TyG = triglyceride-glucose index; BMI = body mass index; RFM = relative fat mass index; LAP = lipid accumulation product index; ascvd-10y-frs = risk of ASCVD 2008 (model with lipid labs); ascvd-10y-frs-simple = risk of ASCVD 2008 (model with BMI).

both clusters 1 and 2 were comparable in terms of their HbA1c and ASCVD risk scores (p-value >0.05,).

## Cardiometabolic profile of female patients with T2DM

The cardiometabolic profile of the population with T2DM is shown in **Table 4**. Among female patients with T2DM, Cluster 3 was distinctively characterized by an excess prevalence of general (80.3%) and central (100.0%) obesity than Clusters 1 and 2. Cluster 2 had a distinct characteristic of a high prevalence of poor glycemic control (96.1%), severe insulin resistance (78.9%), and hypertriglyceridemia (39.5%) compared to clusters 1 and 3. The prevalence of MetS was higher in Clusters 2 (60.5%) and 3 (61.8%) compared to Cluster 1 (24.4%). Furthermore, higher proportions of Cluster 2 than of Clusters 3 and 1 had high phenotype of CVD (p-value = 0.001). The prevalence of metabolic dyslipidemia was 10.5% in Cluster 2 but was absent in Clusters 1 and 3 (**Table 4**).

**Fig 2** provides a visual representation of the different cardiometabolic phenotypes present in patients with T2DM. For example, in cluster 1 of female patients with T2DM, we can observe two phenotypes: normal weight with normal cardiometabolic profile and overweight with poor glycemic control, but a normal cardiometabolic profile. In Cluster 2 of female patients with T2DM, two phenotypes are obvious: obesity with a poor cardiometabolic profile and a high risk of CVD incidence and overweight with a poor cardiometabolic profile but low risk of CVD incidence. In Cluster 3 of women with T2DM, we can observe obese individuals with controlled insulin resistance with or without poor glycemic status and low CVD risk.

## Cardiometabolic profile of male patients with T2DM

Among male patients with T2DM (**Table 4**), cluster 2 was distinctively characterized by a higher prevalence of hypertriglyceridemia, general obesity, central obesity, and severe insulin resistance (17.9%, 28.2%, 79.5% and 28.2%, respectively) than cluster 1 (4.7%, 0%, 7.8% and 7.8%, respectively). Cluster 1 was characterised by a higher prevalence of hypoalphalipoproteinemia

**Table 4. Cardiometabolic risk profile of male and female patients with T2DM.**

| Variable | Male (n = 103) | | | Female (n = 238) | | | |
|---|---|---|---|---|---|---|---|
| | Cluster 1 N = 64 | Cluster 2 N = 39 | p-value | Cluster 1 N = 86 | Cluster 2 N = 76 | Cluster 3 N = 76 | p-value |
| **Cardiometabolic risk** | | | | | | | |
| General obesity | 0.0 | 28.2% | **<0.001** | 2.3%[a] | 25.0%[b] | 80.3%[c] | **<0.001** |
| Central obesity | 7.8% | 79.5% | **<0.001** | 26.7%[a] | 81.6%[b] | 100.0%[c] | **<0.001** |
| High triglycerides | 4.7% | 17.9% | **0.027** | 1.2%[a] | 39.5%[b] | 5.3%[a] | **<0.001** |
| Low HDL-c | 64.1% | 38.5% | **0.011** | 31.4% | 36.8% | 44.7% | 0.214 |
| Metabolic dyslipidemia | 0.0 | 0.0 | | 0.0 | 10.5% | 0.0 | **<0.001** |
| High blood pressure | 29.7% | 30.8% | 0.908 | 19.8% | 21.1% | 21.1% | 0.973 |
| Poor glycaemic control | 54.7% | 51.3% | 0.737 | 48.8%[a] | 96.1%[b] | 31.6%[a] | **<0.001** |
| Severe insulin resistance | 7.8% | 28.2% | **0.006** | 10.7%[a] | 78.9%[b] | 23.7%[a] | **<0.001** |
| MetS | 14.1% | 10.3% | 0.573 | 24.4% | 60.5% | 61.8% | **<0.001** |
| **ASCVD-10-yr risk** | | | | | | | |
| ≤10% increased risk | 73.4% | 71.8% | 0.860 | 100.0 | 100.0 | 100.0 | n/a |
| 10%-20% increased risk | 18.8% | 28.2% | 0.269 | 0.0 | 0.0 | 0.0 | |
| >20% increased risk | 7.8% | 0.0 | 0.075 | 0.0 | 0.0 | 0.0 | |
| **ASCVD-10-yr risk$** | | | | | | | 0.521 |
| ≤10% increased risk | 31.1% | 20.5% | 0.243 | 59.3% | 61.8% | 57.3% | |
| 10%-20% increased risk | 32.8% | 43.6% | 0.273 | 34.9% | 35.5% | 41.3% | |
| >20% increased risk | 35.9% | 35.9% | 1.00 | 5.8% | 2.6% | 1.3% | |
| **CVD risk category** | | | | | | | |
| Low | 59.4% | 28.2% | **0.002** | 80.2%[a] | 59.2%[b] | 77.3%[a] | |
| Intermediate | 20.3% | 46.2% | **0.006** | 8.1% | 1.3% | 4.0% | |
| High | 20.3% | 25.6% | 0.533 | 11.6%[a] | 39.5%[b] | 18.7%[a] | **<0.001** |

MetS metabolic syndrome; ASCVD risk = Framingham 2008 10-year ASCVD risk (model with lipid labs); ASCVD-10-year risk$ = Framingham 2008 10-year ASCVD risk (model with BMI). Superscript alphabets (a, b, c) represent differences between the three cluster according to multiple testing with Bonferroni's correction. Clusters with the same superscript alphabet do not differ significantly.

(64.1% vs. 38.5%) than cluster 2. Metabolic dyslipidemia was absent in both groups, while equivalent proportions of groups 1 and 2 had high blood pressure, poor glycemic control, metabolic syndrome, and high risk of cardiovascular disease. Furthermore, higher proportions of Cluster 2 than of Cluster 1 had an intermediate risk of cardiovascular disease (p-value = 0.006).

As shown in Fig 3, Cluster 1 of male patients with T2DM, three phenotypes were observed: individuals with normal cardiometabolic and low CVD risk profile; those with poor glycemic control but normal cardiometabolic profile, those with poor glycemic control with intermediate-high risk of CVD incidence. Within cluster 2 of male patients with T2DM, we observe three phenotypes: patients with overweight/obesity, severe insulin resistance, poor glycemic control, and high risk of CVD incidence; overweight/obesity patients with poor glycemic control and intermediate risk of CVD; and overweight/obesity patients with normal cardiometabolic profile and low CVD risk profile (**Fig 3**).

Table 5 shows the association between cardiometabolic and cardiovascular risk outcomes and T2DM clusters. Among female patients with T2DM, Cluster 2 was associated with increased high risk of CVD (OR = 5.34, 95% CI: 2.11–13.55) and MetS (OR = 7.07; 95%CI: 3.24–15.42). Furthermore, cluster 3 was significantly associated with an increased risk of developing a MetS.

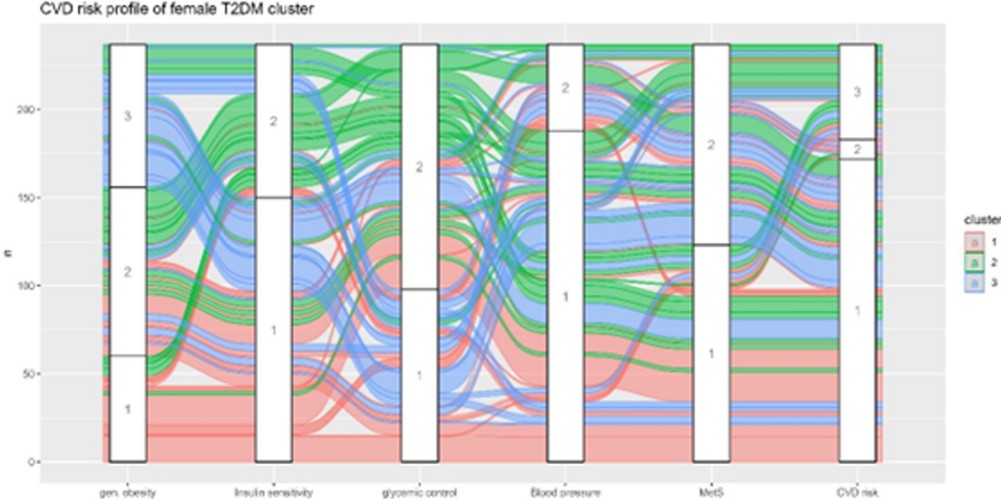

**Fig 2. Spatial view of the cardiometabolic and cardiovascular risk profile of the female T2DM clusters.** Gen obesity: 1 = normal weight, 2 = overweight, 3 = obese; insulin resistance: 1 = optimal, 2 = severe; glycemic control: 1 = normal, 2 = poor; blood pressure: 1 = normal, 2 = high; MetS: 1 = absent, 2 = present; CVD risk: 1 = low, 2 = intermediate, 3 = high.

Among male patients with T2DM, Cluster 2 was also associated with increased intermediate (OR = 21.78, 95% CI 4.17–113.78) and high risk of CVD (OR = 6.84, 95% CI 1.455–32.116), but not MetS and high blood pressure.

## Discussion

In this study, we have characterised subgroups of T2DM marked by obesity, uncontrolled insulin resistance, and poor glycemic control, similar to what has been previously described [20, 63–65]. Therefore, we identified three distinct phenotypes among female patients with

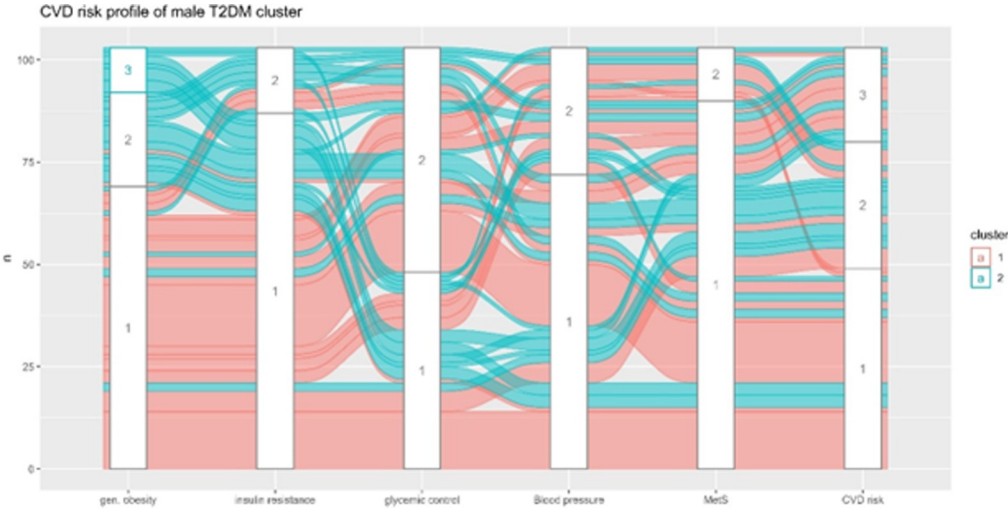

**Fig 3. Spatial view of the cardiometabolic and cardiovascular risk profile of the T2DM male clusters.** Gen obesity: 1 = normal weight, 2 = overweight, 3 = obese; insulin resistance: 1 = optimal, 2 = severe; glycemic control: 1 = normal, 2 = poor; blood pressure: 1 = normal, 2 = high; MetS: 1 = absent, 2 = present; CVD risk: 1 = low, 2 = intermediate, 3 = high, 4 = very high.

**Table 5. Regression analysis showing the association between CVD risk outcomes, T2DM clusters, and other covariates.**

| Risk Outcomes | Variables | P-value | Exp(B) | Lower Bound | Upper Bound |
|---|---|---|---|---|---|
| **Female T2DM patients** | | | | | |
| Intermediate CVD risk^ | Age | <0.001 | 1.28 | 1.12 | 1.46 |
| | Cluster 2 | 0.402 | 0.39 | 0.04 | 3.59 |
| | Cluster 3 | 0.818 | 0.83 | 0.18 | 3.95 |
| High CVD risk^ | Age | <0.001 | 1.12 | 1.06 | 1.18 |
| | Cluster 2 | <0.001 | 5.34 | 2.11 | 13.55 |
| | Cluster 3 | 0.204 | 1.88 | 0.71 | 5.00 |
| MetS# | Age | <0.001 | 1.08 | 1.038 | 1.124 |
| | Cluster 2 | <0.001 | 7.066 | 3.237 | 15.424 |
| | Cluster 3 | <0.001 | 5.944 | 2.755 | 12.823 |
| High blood pressure# | Age | 0.001 | 1.079 | 1.03 | 1.13 |
| **Male T2DM Patients** | | | | | |
| Intermediate CVD risk^ | Age | <0.001 | 1.339 | 1.185 | 1.514 |
| | Cluster 2 | <0.001 | 21.775 | 4.167 | 113.783 |
| High CVD risk^ | Age | <0.001 | 1.229 | 1.104 | 1.369 |
| | Cluster 2 | 0.015 | 6.836 | 1.455 | 32.116 |
| MetS# | Insulin therapy | 0.045 | 12.325 | 1.061 | 143.155 |
| High blood pressure# | Alcohol intake | 0.034 | 4.226 | 1.115 | 16.02 |

All regression models included prescription of medications, patient age, duration of disease, smoking, and alcohol intake status, family history of diabetes and T2DM groups. The symbols represent multinominal regression analysis and # represent logistic regression analysis. In all model T2DM clusters were treated as an entry term, whereas all other variables were conditioned to enter the model using the stepwise forward conditional method.

T2DM: obesity-related phenotype with intermediate CVD risk, severe insulin resistance phenotype with high CVD risk, and 'normal weight-improved insulin resistance' phenotype with low CVD risk. Within the female clusters, there was marked heterogeneity in terms of glycemic control, blood pressure, and metabolic syndrome status. Like the female T2DM patients, two phenotypes were observed in the male population: obesity-related phenotype with severe insulin resistance; and normal weight phenotype with improved insulin sensitivity. Like the female clusters, heterogeneity was observed in the male clusters in terms of glycaemic control, MetS and CVD risk.

Therefore, the T2DM subgroups observed in our study overlap most of the clinically relevant subgroups identified among newly diagnosed T2DM patients [3, 13, 19–21]. In a recent publication including 541 Ghanaians with adult-onset diabetes mellitus, subgroups including obesity-related (73%), insulin-resistant (5%) and insulin-deficient (7%), age-related (10%) and autoimmune-related (5%) were reported [26]. Similar phenotypes including the severe insulin resistance, severe insulin-deficient and obesity-related phenotypes have been replicated using T2DM clinical data obtained from North America, Canada, and Europe [22–24].

The emphasis of many of the recently proposed clustering algorithms for T2DM data assumes that individual patients present a combination of defects in different metabolic pathways, placing them on a quantitative spectrum of metabolic disturbances. Using data from the Outcome Reduction with the Initial Glargine Intervention (ORIGIN) trial [63], Individual Support & Resources for Diabetes (INSPIRED) study [66] and Swedish All New Diabetics in Scania cohort [13], five replicable clusters have been defined for newly diagnosed patients with T2DM: severe autoimmune diabetes, severe insulin-deficient diabetes, severe insulin resistant diabetes, mild obesity-related diabetes and mild age-related diabetes. These classifications

bring out somewhat measurable differences in clinical features between patients and have promising benefits for the practice of precision medicine and clinical research. Using these features in a trial, Dennis *et al.*, [64] reported that precision medicine in T2DM will have a broader clinical utility if it is based on an approach of using specific phenotypic measures to predict specific outcomes. Similarly Bancks *et al.*, [67] employed these subgrouping on a secondary analysis of clinical data and presented the benefits to select intensive lifestyle interventions for patients.

From the pool of literature [65] it is uncertain, which variables and algorithms are appropriate to generate clinically significant T2DM subtypes. It is also uncertain whether the subgroupings identified for newly diagnosed patients will be applicable in patients with longer duration of diabetes. Furthermore, to what extent these groups can be applicable to predict the prognosis in patients with a longer duration of diabetes, and the chances that patients can move from one group to another in the progression of T2DM are unknown.

One thing is certain however, that β-Cell function decline, loss of glycemic control, and weight gain are inherent progressive abnormalities in T2DM, even with treatment [68–71]. Our findings reflect these observations by presenting that the individual with T2DM who has the obesity phenotype and severe insulin resistance phenotype, has a substantial risk of MetS and CVD events. In the report from the ORIGIN trial [63] patients with severe insulin-resistant diabetes subtype presented with a higher incidence of chronic kidney disease and macroalbuminuria, but also benefited more from receiving glargine. Similarly, Xiong *et al.*, [72] reported that patient with a severe insulin-resistant diabetes subtype have increased risk of diabetic retinopathy, diabetic peripheral neuropathy, and hypertension. These findings, together with our present report, indicate that worse outcomes in T2DM may be associated with progressive β-Cell failure not meeting the increasing demand of insulin resistance. It also emphasizes the logic to consider insulin secretion and sensitivity indices in the routine clinical classification and treatment decisions of patients with T2DM.

One of the strengths of the study is that the variables used for clustering were chosen based on their clinical utility, potential pathophysiological proximity, and previous use. The primary limitation is the sample size, which reduces adequate power to back the comparisons between the subgroups and the generalizability of the findings. Another limitation is the varied treatment regimens received by the study enrolled patients which may have a significant effect on the measured parameters. However, none of the participants were receiving SGLT2 inhibitors or GLP-1 agonists, which have shown cardiometabolic benefits in several studies. We also did not consider physical activity, which is a significant modifier of Cardiometabolic risk. Additionally, the potential association of the clusters with CVD incidence was not explored. Although findings from this study are based on observation, it begs the discourse to be validated with more robust designs.

## Conclusion

The study suggests significant cardiometabolic heterogeneity among T2DM patients with differential risk profiles for CVDs. It also highlights the importance of considering insulin secretion and sensitivity indices in clinical decisions regarding patients with T2DM. Thus, clinicians must consider individualized care and treatment options based on the patient's specific cardiometabolic profile which may help prevent future CVD events. Overall, our findings underscores the need for continued research and the development of effective treatment strategies for patients with T2DM.

## Supporting information

**S1 Fig. Boxplots showing the distribution of the data used for the clustering analysis.** Data for female T2DM (left); data for male T2DM (right).
(TIF)

**S2 Fig. Bar graph showing the optimal number of clusters that exist in male and female data according to the majority rule.**
(TIF)

**S3 Fig. A 2D PCA plot showing how the optimal clusters within the male and female T2DM dataset.**
(TIF)

**S1 File. Internal validation estimates of optimal clusters.**
(DOCX)

## Acknowledgments

We would like to thank the staff at the Eastern Regional Hospital, Koforidua, diabetes clinic for their support.

## Author Contributions

**Conceptualization:** Christian Obirikorang, Evans Asamoah Adu, Anthony Afum-Adjei Awuah, Samuel Nkansah Darko, Frank Naku Ghartey, Samuel Ametepe, Eric N. Y. Nyarko, Enoch Odame Anto, William Kwame Boakye Ansah Owiredu.

**Data curation:** Christian Obirikorang, Evans Asamoah Adu, Samuel Ametepe, Enoch Odame Anto.

**Formal analysis:** Evans Asamoah Adu, Enoch Odame Anto.

**Methodology:** Frank Naku Ghartey, Eric N. Y. Nyarko.

**Project administration:** Anthony Afum-Adjei Awuah.

**Supervision:** Samuel Nkansah Darko.

**Writing – original draft:** Christian Obirikorang, Evans Asamoah Adu, Anthony Afum-Adjei Awuah, Samuel Nkansah Darko, Frank Naku Ghartey, Samuel Ametepe, Eric N. Y. Nyarko, Enoch Odame Anto, William Kwame Boakye Ansah Owiredu.

**Writing – review & editing:** Christian Obirikorang, Evans Asamoah Adu, Anthony Afum-Adjei Awuah, Samuel Nkansah Darko, Frank Naku Ghartey, Samuel Ametepe, Eric N. Y. Nyarko, Enoch Odame Anto, William Kwame Boakye Ansah Owiredu.

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
