## [Decision Letter · Decision Letter 0]

1 Sep 2024

PONE-D-24-14354Differential risk of cardiovascular complications in patients with adult type-2 diabetes mellitus in Ghana using clustering analysis: A hospital-based cross-sectional studyPLOS ONE

Dear Dr. Obirikorang,

Thank you for submitting your manuscript to PLOS ONE. After careful consideration, we feel that it has merit but does not fully meet PLOS ONE’s publication criteria as it currently stands. Therefore, we invite you to submit a revised version of the manuscript that addresses the points raised during the review process.

We look forward to receiving your revised manuscript.

Kind regards,

Mohammad Reza Mahmoodi, Ph.D.

Academic Editor

PLOS ONE

Additional Editor Comments:

Corresponding author should be improved English language in revised article as understandable, accurate, and unequivocal. Any typographical or grammatical mistakes and fallacies should be corrected at revision, so please note any specific errors here.

Reviewers' comments:

Reviewer's Responses to Questions

**Comments to the Author**

1. Is the manuscript technically sound, and do the data support the conclusions?

Reviewer #1: Yes

Reviewer #2: Yes

2. Has the statistical analysis been performed appropriately and rigorously? 

Reviewer #1: Yes

Reviewer #2: Yes

3. Have the authors made all data underlying the findings in their manuscript fully available?

Reviewer #1: Yes

Reviewer #2: Yes

4. Is the manuscript presented in an intelligible fashion and written in standard English?

Reviewer #1: No

Reviewer #2: No

5. Review Comments to the Author

Reviewer #1: Journal: The Journal of Clinical Endocrinology & Metabolism

Manuscript ID: PONE-D-24-14354

Title: "Differential risk of cardiovascular complications in patients with adult type-2 diabetes mellitus in Ghana using clustering analysis: A hospital-based cross-sectional study "

Author: Christian Obirikorang et al.

The authors of the present study aimed to identify clinically relevant subgroups of patients with type 2 diabetes mellitus (T2D) by analyzing adiposity, insulin secretion, and resistance indices. Through unsupervised clustering and regression analysis, they distinguished several phenotypes in both female and male patients: for females, the obesity-related phenotype, the severe insulin resistance phenotype, and the normal weight phenotype with improved insulin resistance; and for males, the obesity-related phenotype with severe insulin resistance and the normal weight phenotype with improved insulin sensitivity. The severe insulin resistance phenotype in females was linked to high cardiovascular disease (CVD) risk and metabolic syndrome, while the obesity-related phenotype with severe insulin resistance in males was associated with intermediate and high-risk CVD. The following points should be considered:

Comments:

1. According to the authors, “The findings suggest that there are specific subgroups of patients with T2DM characterized by obesity and uncontrolled insulin resistance leading to poor glycemic control.” Can the authors clarify how this causal assumption is supported by the current analyses?

2. The authors, in the abstract and throughout the manuscript, mention that they have studied " long-term patients with T2DM." Could you please clarify how the long-term characterization is supported given the study population characteristics and the inclusion and exclusion criteria?

3. Please ensure the manuscript defines all abbreviations the first time they are used in the main text and abstract (e.g., MetS). Also, make sure that all abbreviations are defined in each table and figure (e.g., FBS). In addition, if an abbreviation is already defined, you can use it in the text that follows it (e.g., change "Metabolic syndrome" to "MetS" in the “Cluster Analysis” section).

4. The introduction section should end with a discussion of the gap(s) in the literature that this study aims to address, followed by a paragraph(s) highlighting the purpose(s) and aims of this study. The paragraph describing the indices should be discussed earlier.

5. In the “Anthropometry and Blood Pressure Measurements” section, mention if the samples taken were in a fasting state.

6. Please add the relevant references to the text “2013 American College of Cardiology (ACC) and the American Heart Association (AHA) guidelines.”

7. Please add the relevant units for central obesity, if applicable, and a brief comment on the method used or how it was defined.

8. According to the authors, “the World Health Organization’s criteria [3].” Can the authors confirm if reference #3 is appropriate for the WHO statement? Please make sure all the citations in your paper are correct.

9. Please rephrase “Details of the clustering analysis have been attached as separate HTML and Word documents (T2DM_cluster.html and T2DM_analysis.docx).” to a more suitable format for publication and provide a direct way for readers to identify the relevant information (e.g., specify if it is a supplementary note or table). Additionally, I was not able to locate the “T2DM_cluster.html” and “T2DM_analysis.docx” information.

10. Regarding the comment “We tested several methods for an optimal number of clusters,” can the authors provide further information and clarification?

11. I would suggest adding the “Anthropometric indices” section at the beginning of the tables.

12. Did the authors consider the potential role of physical activity in their analyses?

13. Can the authors clarify how the statement “At this point, the development of cardiovascular complications is inevitable” in the Material and Methods section is supported? Also, it seems that the Material and Methods section is not the best place to add this comment. Likewise, for the statement “The combined impact of these abnormalities imposes glucolipotoxicity leading to eventual β-cell decline with decreased insulin secretion,” the Material and Methods section is not the most appropriate section. Please consider moving these statements to the Introduction or Discussion section based on the context.

14. Can the authors clarify what “uncontrolled” insulin resistance phenotype means? How is "uncontrolled" defined?

15. Please check the manuscript for typos and phrasing errors (e.g., “Sulphonyl urea”, "It begs the discourse").

16. In the limitations paragraph, the authors should clearly outline the limitations due to the nature of this study, including generalizability and the fact that the potential association of these clusters with CVD incidence was not explored. Additionally, it should be noted that none of the participants were receiving SGLT2 inhibitors or GLP-1 agonists, which have shown cardiometabolic benefits in several studies.

17. I suggest that the conclusions of the manuscript should also highlight, in a concise manner, the findings of the current study and ensure that the statements closely align with the results of this study.

Reviewer #2: In the present study entitled "Differential risk of cardiovascular complications in patients with adult type-2 diabetes mellitus in Ghana using clustering analysis: A hospital-based cross-sectional study" the authors have reported that there are specific subgroups of patients with T2DM characterized by obesity and uncontrolled insulin resistance leading to poor glycemic control. This underscores the importance of considering differences in adiposity, insulin secretion, and sensitivity indices when making clinical decisions for patients with T2DM.

My comments are as follows:

1. Please change the title of manuscript to "Differential risk of cardiovascular complications in patients with type-2 diabetes mellitus in Ghana: A hospital-based cross-sectional study

2. Please write keywords as follow:

Type-2 diabetes, cluster analysis, cardiovascular risk, odds ratio

3. Introduction is too long, this section needs to be summarized.

4. Gap of knowledge and the novelty of study should be added in introduction section.

5. The aim of study should be clearly added in the last paragraph of introduction.

6. The resolution of figures 1-3 should be increased.

7. Tables should be revised. Please delete the lines into tables.

6. PLOS authors have the option to publish the peer review history of their article (what does this mean?). If published, this will include your full peer review and any attached files.

Reviewer #1: No

Reviewer #2: No

---

## [Author Response · Author response to Decision Letter 0]

4 Oct 2024

Journal: PLOS ONE

Ref: Submission ID PONE-D-24-1435

Re: Differential risk of cardiovascular complications in patients with adult type-2 diabetes mellitus in Ghana using clustering analysis: A hospital-based cross-sectional study"

Corresponding Author 

Christian Obirikorang, BSc, PhD

Kwame Nkrumah University of Science and Technology

Kumasi, GHANA

krisobiri@yahoo.com

Re: Response to Reviewer Comments

Dear Editor, 

We would like to thank you and the reviewers for the constructive feedback to improve the quality of the manuscript. Please find below our point-by-point response to the comments raised by the reviewers. 

Reviewer reports

 Reviewer #1

Comment: According to the authors, "The findings suggest that there are specific subgroups of patients with T2DM characterized by obesity and uncontrolled insulin resistance leading to poor glycemic control." Can the authors clarify how this causal assumption is supported by the current analyses?

Response: Thank you for the comment. This statement was inferred from the Alluvial plot (Figure 2 and 3) which shows the patterns of cardiometabolic risk profile of the participants at the individual level. A distinctive cluster of patients presented with concurrent obesity and sever insulin resistance also had poor glycaemic control. In addition to Figure 2 and 3, Tables 3 and 4 highlighted that the clusters with higher proportion of obesity and insulin resistance also presented with higher proportions of poor glycaemic control. 

Comment: The authors, in the abstract and throughout the manuscript, mention that they have studied " long-term patients with T2DM." Could you please clarify how the long-term characterization is supported given the study population characteristics and the inclusion and exclusion criteria?

Response: The authors are grateful for the comment. We used long-term to refer to T2DM population that has been on treatment for at least a year and are still on treatment with regular care at the time of the study. We have described this in the methodology section under sample design and population as well as the inclusion and exclusion criteria. 

Comment: Please ensure the manuscript defines all abbreviations the first time they are used in the main text and abstract (e.g., MetS). Also, make sure that all abbreviations are defined in each table and figure (e.g., FBS). In addition, if an abbreviation is already defined, you can use it in the text that follows it (e.g., change "Metabolic syndrome" to "MetS" in the "Cluster Analysis" section).

Response: Thank you for the comment. We have taken note of this and carefully corrected it in the revised submission. 

Comment: The introduction section should end with a discussion of the gap(s) in the literature that this study aims to address, followed by a paragraph(s) highlighting the purpose(s) and aims of this study. The paragraph describing the indices should be discussed earlier.

Response: Thank you for the comment. We have introduced several lines of discussion on the recent gaps in literature that the study aims to address. In the first paragraph, we highlighted that the current approach to classifying T2DM falls short in understanding the epidemiological and clinical landscape of the disease. In the third paragraph, we emphasized on the mounting responsibility of managing diabetes and the shortcomings of the current standard definition. In the fourth paragraph, we introduced the potential of unsupervised learning models to overcome these gaps. In the last paragraph, we have introduced the main aim of the study including the summary measures, primary outcomes the context of the objective. Additionally, we have moved the discussion of the indices to the methods under “parameter” that gives a thorough justification of the selected indices for the clustering algorithm. 

Comment: In the "Anthropometry and Blood Pressure Measurements" section, mention if the samples taken were in a fasting state.

Response: Thank you for the comment. We have introduced under data and sample collection section that “It is routine practice for patients to fast overnight before visiting the clinic for health examinations. Thus, laboratory measurements are taken in fasting state”. Additionally, we have “All measurement were performed whiles patients were in the fasting state” to the Anthropometry and Blood Pressure Measurements section. 

Comment: Please add the relevant references to the text "2013 American College of Cardiology (ACC) and the American Heart Association (AHA) guidelines."

Response: Thank you for the comment. We have added the associated reference (ref: #30) by Stone et al, 2014. 

Comment: Please add the relevant units for central obesity, if applicable, and a brief comment on the method used or how it was defined.

Response: Thank you for the comment. Central obesity was defined using relative fat mass index (RFM) thresholds, which is a derived index from waist circumference and height. Thus, it has no unit. Under the “Definition of other outcome variables” section we have provided details on how central obesity was defined, which based on sex and age-specific cut-off values for RFM. 

Comment: According to the authors, "the World Health Organization's criteria [3]." Can the authors confirm if reference #3 is appropriate for the WHO statement? Please make sure all the citations in your paper are correct.

Response: Thank you for the comment. We have corrected the reference and cited the appropriate source (ref: #31). 

Comment: Please rephrase "Details of the clustering analysis have been attached as separate HTML and Word documents (T2DM_cluster.html and T2DM_analysis.docx)." to a more suitable format for publication and provide a direct way for readers to identify the relevant information (e.g., specify if it is a supplementary note or table). Additionally, I was not able to locate the "T2DM_cluster.html" and "T2DM_analysis.docx" information.

Response: Thank you for the comment. The statement has been revised to “The details of the clustering analysis can be found in the supplementary file 1”. We have attached two supplementary files. File 1 outlines all the steps taken to perform the clustering analysis including data and comments which was not presented in the main manuscript. File 2 contains additional data in the form of Tables and Figures. 

Comment: Regarding the comment "We tested several methods for an optimal number of clusters," can the authors provide further information and clarification?

Response: The authors are grateful for this comment. We evaluated all available methods for unsupervised clustering to select the best fit for our data. The details of this have been provided in Supplementary file 1. 

Comment: I would suggest adding the "Anthropometric indices" section at the beginning of the tables.

Response: Thank you for this suggestion. We have added "Anthropometric indices" to tables 2 and 3 to reflect the content of the table. 

Comment: Did the authors consider the potential role of physical activity in their analyses?

Response: Thank you for the comment. No, we did not. We have included a statement to address this as a limitation. 

Comment: Can the authors clarify how the statement "At this point, the development of cardiovascular complications is inevitable" in the Material and Methods section is supported? Also, it seems that the Material and Methods section is not the best place to add this comment. Likewise, for the statement "The combined impact of these abnormalities imposes glucolipotoxicity leading to eventual β-cell decline with decreased insulin secretion," the Material and Methods section is not the most appropriate section. Please consider moving these statements to the Introduction or Discussion section based on the context.

Response: Thank you for the comments. We have added appropriate references to support the statement. In the revised manuscript, under “Cluster analysis” we have separated this for clarity. This statement represents our assumptions and context for selecting the variables used for the cluster analysis. Thus, it is more appropriate to keep it in the method section. Kindly see the “Cluster analysis” section of the revised manuscript. 

Comment: Can the authors clarify what "uncontrolled" insulin resistance phenotype means? How is "uncontrolled" defined?

Response: Thank you for the comment. We have corrected this error in the revised manuscript. 

Comment: Please check the manuscript for typos and phrasing errors (e.g., "Sulphonyl urea", "It begs the discourse").

Response: Thank you for the comment. We have carefully checked and corrected all typos. 

Comment: In the limitations paragraph, the authors should clearly outline the limitations due to the nature of this study, including generalizability and the fact that the potential association of these clusters with CVD incidence was not explored. Additionally, it should be noted that none of the participants were receiving SGLT2 inhibitors or GLP-1 agonists, which have shown cardiometabolic benefits in several studies.

Response: Thank you for the comment. The limitation section has been revised to include these suggestions. Kindly see the limitation section in the revised manuscript. 

Comment: I suggest that the conclusions of the manuscript should also highlight, in a concise manner, the findings of the current study and ensure that the statements closely align with the results of this study.

Response: Thank you for the comment. We have revised the conclusion in a more concise manner that reflect the findings of the study. Kindly see the revised submission. 

Reviewer #2

Comment: Please change the title of manuscript to "Differential risk of cardiovascular complications in patients with type-2 diabetes mellitus in Ghana: A hospital-based cross-sectional study

Response: Thank you for the comment. The title has been revised as suggested. 

Comment: Please write keywords as follow:

Type-2 diabetes, cluster analysis, cardiovascular risk, odds ratio

Response: Thank you for the comment. The keywords have been revised as indicated. Kindly see the revised submission. 

Comment: Introduction is too long; this section needs to be summarized.

Response: Thank you for the comment. This has bee addressed in the revised manuscript. Kindly see the introduction section of the revised submission. 

Comment: Gap of knowledge and the novelty of study should be added in introduction section.

Response: Thank you for the comment. We have introduced several lines of discussion on the recent gaps in literature that the study aims to address. In the first paragraph, we highlighted that the current approach to classifying T2DM falls short in understanding the epidemiological and clinical landscape of the disease. In the third paragraph, we emphasized on the mounting responsibility of managing diabetes and the shortcomings of the current standard definition. In the fourth paragraph, we introduced the potential of unsupervised learning models to overcome these gaps. In the last paragraph, we have introduced the main aim of the study including the summary measures, primary outcomes the context of the objective

Comment: The aim of study should be clearly added in the last paragraph of introduction.

Response: Thank you for the comment. We have added the aim of the study as suggested. Kindly see the last paragraph in the introduction section of our revised submission. 

Comment: The resolution of figures 1-3 should be increased.

Response: The current resolution of the figure is increased to 600 dpi. 

Comment: Tables should be revised. Please delete the lines into tables.

Response: Thank you for the comment. This has been adequately addressed in the revised manu

---

## [Decision Letter · Decision Letter 1]

1 Nov 2024

Differential risk of cardiovascular complications in patients with adult type-2 diabetes mellitus in Ghana using clustering analysis: A hospital-based cross-sectional study

PONE-D-24-14354R1

Dear Dr. Obirikorang,

We’re pleased to inform you that your manuscript has been judged scientifically suitable for publication and will be formally accepted for publication once it meets all outstanding technical requirements.

Kind regards,

Mohammad Reza Mahmoodi, Ph.D.

Academic Editor

PLOS ONE

Additional Editor Comments (optional):

Reviewers' comments:

Reviewer's Responses to Questions

**Comments to the Author**

1. If the authors have adequately addressed your comments raised in a previous round of review and you feel that this manuscript is now acceptable for publication, you may indicate that here to bypass the “Comments to the Author” section, enter your conflict of interest statement in the “Confidential to Editor” section, and submit your "Accept" recommendation.

Reviewer #1: All comments have been addressed

Reviewer #2: All comments have been addressed

2. Is the manuscript technically sound, and do the data support the conclusions?

Reviewer #1: Yes

Reviewer #2: Yes

3. Has the statistical analysis been performed appropriately and rigorously? 

Reviewer #1: Yes

Reviewer #2: Yes

4. Have the authors made all data underlying the findings in their manuscript fully available?

Reviewer #1: Yes

Reviewer #2: Yes

5. Is the manuscript presented in an intelligible fashion and written in standard English?

Reviewer #1: Yes

Reviewer #2: Yes

6. Review Comments to the Author

Reviewer #1: Journal: The Journal of Clinical Endocrinology & Metabolism

Manuscript ID: PONE-D-24-14354

Title: " Differential risk of cardiovascular complications in patients with type-2 diabetes mellitus in Ghana: A hospital-based cross-sectional study"

Author: Christian Obirikorang et al.

The authors have satisfactorily responded to my comments and suggestions, and their revisions have further improved the quality of the paper. I have no further comments.

Reviewer #2: All comments have been addressed.

7. PLOS authors have the option to publish the peer review history of their article (what does this mean?). If published, this will include your full peer review and any attached files.

Reviewer #1: No

Reviewer #2: No

---

## [Editor Report · Acceptance letter]

14 Nov 2024

PONE-D-24-14354R1 

PLOS ONE

Dear Dr. Obirikorang, 

I'm pleased to inform you that your manuscript has been deemed suitable for publication in PLOS ONE. Congratulations! Your manuscript is now being handed over to our production team.

Kind regards, 

on behalf of

Dr. Mohammad Reza Mahmoodi 

Academic Editor

PLOS ONE